# Latino/a experiences of homelessness in California: Qualitative findings from the California Statewide Study of People Experiencing Homelessness (CASPEH)

Zena K. Coronado[1,2], Michael Duke[1,2], Madison Rodriguez[1,2], Lourdes Johanna Avelar Portillo[1,2], Dafna Erana Hernandez[1,2], Margot Kushel[1,2]*

1 Division of Health and Society, Department of Medicine, University of California, San Francisco, California, United States of America, 2 Benioff Homelessness and Housing Initiative, University of California, San Francisco, California, United States of America

* margot.kushel@ucsf.edu

## Abstract

The number of Latino/a people experiencing homelessness in the United States is increasing due to increased housing costs, economic disruption from the COVID-19 pandemic, and increased homelessness among migrants. There is little known about the lived experiences of Latino/a individuals experiencing homelessness. We conducted qualitative interviews in English and Spanish with 84 participants who self-identified as Latino/a as part of a large mixed methods representative study of homelessness. We analyzed qualitative interview transcripts from the California Statewide Study of People Experiencing Homelessness (CASPEH) to elucidate factors contributing to increased risk of homelessness for the Latino/a population, the vulnerabilities Latino/a individuals face while navigating homelessness, and how these vulnerabilities influence access to housing and services. Latino/a participants reported numerous factors that precipitated their descent into homelessness and challenged their ability to access social services and other resources. These factors include: 1) limited familial and social support, 2) barriers to housing returns, and 3) job loss precipitating homelessness and ongoing barriers to employment. We defined housing returns as successfully regaining housing such that participants no longer met criteria for homelessness. This could be through renting their own room or apartment, moving in with family or friends long-term, or moving into subsidized housing or permanent supportive housing. Our findings highlight challenges impacting Latino/a adults experiencing homelessness in California and provide evidence for developing culturally centered and programmatic interventions to address homelessness among this population.

**Data availability statement:** Because of the high likelihood of respondents describing illegal or socially-sanctioned behaviors in specific community settings, coupled with the current targeting of immigrant populations for deportation, our Lived Expertise Advisory Board urged us to maintain control of the de-identified data. While we collected the data anonymously, the detailed data given by participants in in-depth interviews (which often included information such as documentation status, location of interview, and detailed information that could possibly be identifying), the Community Advisory Board asked us to exercise caution. In light of these factors, and in consultation with UCSF's Director of Data Science & Open Scholarship, we agree to share the study data based on individual requests to the UCSF Office of Research at research@ucsf.edu.

**Funding:** This research was supported by the California Healthcare Foundation (G 31355), the Blue Shield of California Foundation (RP-2011-15087 and COV-2206-18206), and the University of California, San Francisco Benioff Homelessness and Housing Initiative (BHHI). The funders had no role in study design, data collection, data analysis, decision to publish or preparation of the manuscript.

**Competing interests:** The authors have no financial or non-financial interests to disclose.

## Introduction

The number of people experiencing homelessness in the United States who identify as Latino/a increased by 38% between 2019 and 2023, due to increased housing costs, economic disruption from the COVID pandemic, and displacement of people who had migrated to the United States [1]. The Latino/a population in the U.S. is diverse with regard to citizenship status, language capabilities, and country of origin; these result in differential risk of and pathways to homelessness. A combination of socioeconomic, institutional, and immigration-related barriers and experiences of discrimination elevates the risk of homelessness among the Latino/a population.

Latino/a people face higher poverty rates and lower income and employment opportunities than white Americans [2,3]. They are more likely to be extremely low-income and experience housing cost-burden [4–6]. Hiring discrimination and bias against foreign credentials impair the economic outcomes of even the most educated Latino/a people, especially those who are foreign-born [7,8]. Undocumented legal status contributes to the risk of homelessness by impairing income opportunities, restricting access to safety-net protections, and limiting options to build credit or rental histories [3,9]. Compounding these economic stressors is the severe shortage of affordable housing in the U.S., with only 35 units per 100 extremely low-income households nationwide, and just 24 per 100 in California [4,5,10]. Together, these conditions increase the risk of housing problems in Latino/a populations, including high housing cost burden, living in overcrowded or subpar housing, doubling up with family or friends without rental protection, or experiencing homelessness, defined by the Federal Homeless Emergency Assistance and Rapid Transitions to Housing (HEARTH) Act, as living in a shelter or in a place not meant for human habitation (vehicle, outdoors, abandoned building) [11].

The economic and housing market effects of the COVID-19 pandemic worsened the housing conditions of Latino/a populations and contributed to the recent rise in Latino/a homelessness [12,13]. The pandemic disproportionately impacted Latino/a communities across the U.S. and widened pre-existing racial and ethnic disparities [14,15]. Both living in crowded households and being employed as essential or food service workers, common in Latino/a populations, increased exposure to the SARS-Cov-2 virus [16]. Unemployment and income loss disproportionately affected Black and Latino/a workers [9,12]. Many undocumented Latino/a immigrants and mixed-status Latino/a families lacked social security numbers and proof of unemployment; thus, they were excluded from federal relief provisions under the Coronavirus Aid, Relief, & Economic Security (CARES) Act [17]. Many of those who qualified for aid were reluctant to participate in these programs due to concerns about the public charge policy that could deny them from obtaining a green card, visa, or citizenship status for requesting aid [14,16,17]. With limited access to these programs, some responded by moving in with family and friends and some entered homelessness [14,17]. These environments render Latino/as who do meet the HEARTH Act definition of homelessness less likely to be included in the Federally mandated Point-in-Time count and in research studies of homelessness that sample from service

venues [18]. Stigma, lack of awareness of resources, and a dearth of linguistic and cultural competency among service providers contributes to the underutilization of services among this population [19,20].

There is little known about the lived experience of homelessness among Latino/a people. Understanding the socio-structural conditions that create the risk for homelessness in the Latino/a population and their unique challenges in accessing housing and social safety net services is important for developing solutions to address Latino/a homelessness. Based on qualitative data from a large-scale representative study of homelessness in California, we explore the distinct and complex factors that shape the lived experiences of Latino/a Californians experiencing homelessness.

## Methods

The California Statewide Study of People Experiencing Homelessness (CASPEH) is the largest representative study of homelessness in the U.S. since the mid-1990s, and the first large-scale study to use a mixed-methods approach (administered questionnaires and paired in-depth interviews). From October 2021 to November 2022, the research team surveyed a representative sample of people experiencing homelessness in eight regions of California (N = 3,200). To recruit participants, we used venue-based sampling supplemented by respondent driven sampling. We recruited participants from a random sample of venues where adults experiencing homelessness congregate (e.g., shelters, community centers, food programs, showers, and unsheltered encampments). We used respondent driven sampling to recruit hard-to-reach populations, including farm workers and day laborers who are overwhelmingly Latino/a [21,22]. The study received approval from the University of California, San Francisco Institutional Review Board (IRB #: 20–33117). Verbal informed consent was obtained from each participant prior to participation in the study. Participants received a verbal explanation of the study's objectives, protocols, potential risks, measures to maintain confidentiality and privacy, and how their data would be used. Eligibility criteria included being at least 18 years of age and homeless, as defined by the HEARTH Act, ability to provide informed consent, and not having a previous record of participating in the CASPEH [23].

To understand the full context of participants' experiences and aid in the interpretation of survey data, we developed seven conceptually overlapping qualitative interview-based sub-studies: barriers to housing returns, behavioral health, precipitants of homelessness, Black experiences of homelessness, Latino/a experiences of homelessness, incarceration, and intimate partner violence. We used purposive sampling to select participants for the qualitative interviews (N = 365). Purposive sampling is a technique in qualitative research to identify participants who have information, experiences, or characteristics that are relevant to the aims of the study.

Participants were selected based on a) having completed the survey b) being flagged for a particular qualitative sub-study based on their responses to specific survey questions, and c) the survey interviewer's assessment of the participant's ability to discuss the topic at length. Qualitative interviews included a series of open-ended questions to elucidate the respondents' experiences of homelessness; each sub-study used a unique interview guide. The Latino/a experiences study included topics on familial and social support, immigration and citizenship, language barriers, and discrimination. We intentionally focused our analysis on themes participants discussed most often, excluding topics that were less frequently mentioned such as healthcare access, mental and behavioral health, and other health-related experiences. This approach aligned with our research objectives and the scope of this manuscript.

We administered the qualitative interview portions in English or Spanish. We provided a $30 gift card for participating in the survey and an additional $30 gift card for completing a qualitative interview. All interviews lasted 30–45 minutes each and were audio-recorded and transcribed. Transcriptionists transcribed the Spanish interviews verbatim in Spanish then translated them to English, and Spanish-speaking staff performed quality assurance on the final English-version transcripts.

This paper uses data from qualitative interviews with participants across the qualitative sub-studies who self-identified as Latino/a (n = 84); 35 participated in the Latino/a sub-study and 51 in one of the other studies. Thirty percent were conducted in Spanish. Among the 84 Latino/a participants, 67% identified as cis-men and 32% as cis-women, and 30% were

born outside the U.S. (see Table 1). We created a single coding manual to allow for analysis both within and across the sub-studies, because they contained substantial thematic overlap. We used a deductive approach to build a thematically broad group of codes based on the research questions for each sub-study; we then added additional codes inductively by closely reading the interview transcripts. By employing a structured coding manual to organize and categorize the qualitative data, we used an interpretivist codebook approach for our thematic analysis [24]. We then used the coded materials to identify emerging themes from the data [25].

We coded and analyzed the qualitative interview transcripts using the Dedoose (version 9.0) software platform. We assigned coding teams, consisting of a primary and secondary coder, to each sub-study. The primary coder initially coded the interview transcript. The secondary coder then reviewed and, when relevant, added to the primary coder's coding decisions. We developed several protocols to ensure intercoder reliability. We report details of our data collection and coding processes elsewhere [26].

## Results

Participants described a variety of experiences of homelessness, reflecting the diversity of legal status, country of origin, language, and characteristics of their location within California (e.g., population density, proportion of Latino/a population within communities). Participants' experiences varied based on whether they were first-generation, monolingual or Spanish-dominant, and/or undocumented. We identified three major themes: 1) limited familial and social support, 2) barriers to housing returns, and 3) job loss precipitating homelessness and barriers to employment. We describe these below.

### Limited familial and social support

**Social networks played a role in supporting participants, but were unable to stave off homelessness.** Latino/a participants relied on their families, friends, and other members of their social network for housing, financial, and other forms of support. When unable to afford the full cost of housing, participants moved in—without formal rental agreements-- with family, friends, or acquaintances ("doubling up"). While these arrangements do not meet the federal definition of homelessness, participants described them as overcrowded, chaotic, and substandard. While these strategies staved off homelessness for the short term, participants' concerns that they would become a financial burden on housed family members and friends limited the length of stay; family or friends, at times, would force participants to leave with little warning.

**Table 1. Sample population demographic characteristics (n = 84).**

| Characteristic | n (%) |
|---|---|
| Gender Identity | |
| Men | 56 (67) |
| Women | 27 (32) |
| Non-Binary/Transgender | 1 (1) |
| Age (years) | |
| 18-24 | 4 (5) |
| 25-44 | 32 (38) |
| 45-64 | 39 (46) |
| 65+ | 9 (11) |
| Born outside the U.S. | |
| Yes | 25 (30) |
| Interview Language | |
| Spanish | 25 (30) |

Participants reported periods of moving between family and friends for brief periods ("couch surfing"), when they did meet the federal definition of homelessness, but did not self-identify as homeless. One participant shared:

*Well, my mom passed away last year. So I was always able to run back to Mom's. I don't have that anymore … my sister will let me stay every so often… a couple days or whatever. But it's never very long. And it's just because of my nephew always just got an attitude… he just thinks I'm a freeloader.*

Some participants reported fluctuating between couch surfing and other forms of homelessness. During their current episode of homelessness, participants occasionally stayed with family or friends, but were more often unsheltered, living outside, in vehicles, or in places not intended for human habitation, including abandoned buildings and unfurnished garages without access to running water or electricity. One participant described:

*I ask if there is an available place for renting. If everything is occupied, you look for cars to sleep in. And ask the owners to let you sleep in [one] for one night or so. They let you stay one night or something like that, and next morning you must leave, you can't stay any longer because they don't want you there. So, you leave and look for another place.*

**Need to support family decreased participants' ability to maintain housing.** Immigrant men whose families remained in their country of origin described their desire to maintain their paternal roles and fulfill responsibilities as head of their household. They tried to spend as little as possible on their own housing and living expenses to send remittances to their families, as described by one participant:

*Sometimes you need to help your family, help your wife in El Salvador, help your daughter and everything for her school, so then I don't have enough money to pay for rent and everything else. And for that reason, I say that I'm going to stay outside because I feel like I live good, and the little bit of money I have is enough for me to send to them and eat here as much as I can.*

These participants reported conflicting priorities. While participants desired housing, family obligations to send money home constrained participants' ability to maintain their own housing and prolonged their experiences of homelessness.

**Barriers to receiving familial support.** Participants described feelings of shame, pride, and independence as barriers to staying with people they know. Some described not wanting to be a burden to their families nearby who were facing economic precarity. Others were reluctant to seek help from their social networks because they felt ashamed that they were falling behind or struggling to survive on their own. This was especially true for participants who identified as men, as one shared:

*I do have family that I could talk to, but I don't really let them, I mean, my situation is not good, and I need to make it better myself. And I don't want to tell people my situation for them to feel sorry for me, you know what I mean? Or I told you so or you know what I mean? I'm at the age where I know what I should be doing and what I should've could've done…*

Participants described being torn because they wanted to be independent from family and friends but relied on them to meet basic needs. Many expressed a strong desire to resolve their housing situation on their own, reflecting culturally prescribed values pertaining to stoicism and self-reliance. One participant explained:

*I'm very closed about what I'm going through. None of my family knows I'm homeless, none of my friends. I just don't want to tell anyone. I want to be able to get out of this alone. It feels like I'm stuck in this, and it sucks.*

### Barriers to housing returns

**Structural barriers and discrimination.**  Participants experienced several barriers to accessing social services and obtaining housing. Most participants described the process of applying for market-rate and subsidized housing and housing subsidies as time-consuming and confusing. Some found it challenging to navigate online housing applications or did not know where to apply.

*I just wish they had more open doors when it comes to housing… the trouble I've had is finding who, what, you know? You hear about Section 8 housing. How do I get in there?... You can fill out an application online, but you never get no phone calls, you could never call them… they say the waiting list is like years and years away. And it's like, man, I really wish there was an easy way… or somewhere where you can go and work with people that can help you find this housing and stuff.*

Others, particularly older Latino/a adults, described limited English language skills as a barrier. One older Spanish speaking participant recounted experiencing a language barrier when trying to access housing services:

*There was a person who was speaking with me, with a lot of shame, but I couldn't manage the language, English… A person was contacting me, I don't know, there are some places that tell you… that the government offers housing for homeless people. But to this day, there are many people waiting. And I don't know and I haven't known what the problem was... but that person gave up because of how we just talked and talked.*

Several participants experienced discrimination based on their ethnic identity, appearance, and homelessness status when trying to obtain housing or services. Some perceived that people of other races and ethnicities received priority over them when trying to access services, particularly those who shared the same racial or ethnic background as the service provider staff.

Experiences of discrimination discouraged participants from seeking services and other assistance. As one participant described:

*I think [race] has to do with it a little bit... my name says it all... 'I'm [first name] Garcia (pseudonym).'... That name is so common that – 'That's another Mexican... Let's put them at the end of the line again.'... That's the type of treatment that I've been getting. Everywhere I go, I'm like this, 'I'm going to go change my name, maybe, you know, Smith.'...It's getting to me where I don't even try anymore...Because it's repetitive.*

Some participants recounted experiencing differential treatment in shelter settings. While some residents were allowed to break shelter rules, like arriving late for curfew, Latino/a participants described facing harsher consequences for doing the same.

*Here [at the shelter], though, yeah, like, sometimes they ignore me, or they give other people special privileges and stuff, you know. Like, they'll let people come and go when they want, and if I come in late, I'll get written up…*

Others described experiences of discrimination from their past and potential neighbors, noting that they did not feel welcomed in their communities.

*There was a woman, I don't know [what was wrong with her] and she came out saying a lot of things and she told the children that if I moved in there (into the apartment complex) she would set our place on fire with everything and the children. For me, I don't know if she was crazy, but that is like discrimination.*

**Police interactions.** Participants described experiences of harassment by police, especially when living outdoors, in vehicles, or in other unsheltered environments. Some perceived that they were subjected to increased police harassment while unhoused because of their Latino/a identity:

*It all depends... because the police is always harassing – 'Oh, you can't park here.' I mean they'll try to find anything to get you out and harass you, especially if you're Hispanic.*

Others described how police interactions created additional challenges around navigating homelessness, noting that these interactions sometimes came with verbal threats of arrest or jail time, and the potential loss of their pets or belongings, as one participant described:

*You get sick of the grind… every time you see a cop car, they're not there to help you…they laugh at us, mock us, call us names, make fun of us. I mean, that day I wasn't moving fast enough and so he (the police officer) kept telling me… 'You need to get, you need to get, get, I'm not letting you take anything else,' he was like, 'You either go now or I'm going to throw you in jail and I'm going to take your dog.'*

**Immigration status and legality.** Participants discussed how their lack of citizenship status created barriers to financial and housing stability. Lacking documentation or citizenship status impeded access to government benefits like the Supplemental Nutrition Assistance Program, Medicaid, unemployment insurance, and Housing Choice Vouchers. Participants sought work permits or permanent residency status, though often unsuccessfully. One undocumented participant described relying on their own resources as a means for survival because they do not qualify for government benefits:

*Wages are not enough around here, especially for me since I am undocumented. I can't apply for benefits like unemployment, I try to take care of my money so that I have enough until I start working again and get funds together to survive. That is the way you survive here.*

Participants whose immigration status allowed for eligibility faced other barriers to housing. Many reported having lost or been unable to renew essential documents needed to secure housing (e.g., green cards or identification cards) due to their homelessness and faced long wait times to receive replacements. One participant shared:

*Yes, I applied for Section 8, and they already gave me the voucher, but I can't go because they want proof of payment, to charge me 30% or something like that, but since I don't have my MICA (Border Crossing Card) [because it expired], and I can't get paid with a check, they don't want to take me in the apartments. So, I need to wait for my proof of payment. The question is, are they (Section 8) going to wait the three months? Because the woman from Social Services called, and they told her it can take up to three months to send the MICA.*

### Job loss precipitating homelessness and barriers to employment

**Exposures to health and safety hazards.** Most men who were Latino/a and employed prior to homelessness worked in construction, landscaping, farmwork, and the service sector. Working in these sectors was more common among those born abroad, as these positions did not require English language fluency or legal documentation. Participants who were born outside the U.S. noted that they were motivated to move in part by these positions paying more in California than similar jobs in their home country.

Workers in these industries, regardless of gender, faced a high risk of injury, a risk made worse because many employers did not provide proper protective equipment. Few of these employers provided health insurance or paid into workers' compensation. As a result, workers who became injured faced losing their job, leaving them vulnerable to homelessness, as described by one participant:

*In my jobs I did feel discrimination… I have had situations where because I am a woman, because I am Latina, In my job I've had to work extra, to do heavy duty work, I gave a lot of effort and because of that I had an accident while I was working in a hotel, and that totally changed my way of life and is why I am in my current situation now.*

Other Latino/a workers reported having been employed in the service sector, working in restaurants, retail establishments, or as delivery drivers. The COVID-19 pandemic placed these workers at risk for both infection and job loss. Some participants reported that they quit their job because they did not feel safe being employed as essential workers due to the risk of contracting COVID-19. Others reported losing their jobs due to COVID-19-related closures. Some participants reported receiving COVID-19 stimulus checks but noted that they did not provide enough to prevent homelessness. Some reported trying to receive stimulus checks, but not getting them, attributing this to their lack of a consistent home address, not having previously submitted income tax forms, or being undocumented, as one participant expressed:

*I have my driver's license, I´ve never had a ticket, I've never had accidents, I'm clean of everything. I've never been in jail; I don't have any felonies. I am just waiting for an opportunity to work and establish my taxes. For that reason, I have not been able to get any support like the one there was with COVID [because I don't have the MICA].*

**Poor health as a barrier to employment.** Participants reported being highly motivated to get a job, even among those for whom advanced age or physical infirmities would make employment difficult. Although Latino/a participants were highly motivated to work, poor health often led to job loss and subsequent homelessness. The longer they remained homeless, the worse their health became, interfering with their ability to work. Physical health, transportation challenges, and the absence or loss of legal documents (e.g., drivers licenses) proved to be substantial barriers to obtaining employment. For example, a 61-year-old man told us:

*I don't want disability. What I want is to work to be able to pay for my housing…If I have disability, I can't work at anything. And I still feel a little good to be able to work.*

Because participants experiencing poor health faced substantial challenges in obtaining employment in the formal sector, they often turned to informal avenues for generating income, particularly recycling. Participants reported spending several hours per day recycling. Despite these efforts, they did not receive enough money to meet their basic needs.

## Discussion

In a large qualitative study of Latino/a adults, recruited purposively from a representative study of adults experiencing homelessness in California, we found that Latino/a participants experienced challenges related to accessing social support networks, underutilized housing and other social services, and faced employment precarity that resulted in homelessness. While people experiencing homelessness who identify as Latino/a have a range of experiences shaped by their immigration experience and status and language capability, they shared a set of experiences that shaped their pathway into and experience of homelessness.

We found that Latino men were hesitant to rely on familial support due to cultural norms that validate self-sufficiency and stoicism. There is substantial literature describing the importance of kin-based networks as sources of mutual aid in Latino/a families, but we found that economic and related factors constrained participants' ability to use these networks,

including concerns about burdening members of their network who themselves faced economic hardship [27–31]. These findings are similar to those of other populations in the United States, including the widespread traits of individualism and self-reliance and the economic precarity faced by low-income households regardless of their racial or ethnic composition. Our findings contribute to the literature by showing the ways in which Latino/a people experiencing homelessness grapple with trying to balance the cultural norms regarding self-reliance and the use of family as a source of support and mutual aid.

We found that Latino/a populations under-utilize services, including housing and homelessness prevention services, supporting nascent research on Latino/a homelessness [19,32–34]. The reasons Latino/a do not access services are complex and differ according to immigration status, language skills, and age [32]. Participants reported that language barriers and the complicated process of completing paperwork made it challenging to access and use services successfully. Participants who accessed housing and homelessness services faced discrimination. Negative and perceived discriminatory encounters with law enforcement officers created additional barriers to safety and housing. The intersectional discrimination that Latino/a people experiencing homelessness face as unhoused people and people of color corresponds with similar forms of prejudice and discrimination among Black populations experiencing homelessness [35]. Our research provides a clear illustration of Olivet and colleagues' contention that "disproportionate rates of homelessness among people of color can be understood as a symptom of the failure of multiple systems to provide equal opportunity for all racial and ethnic groups" [36].

Lastly, our findings illustrate the relationship between loss of employment and loss of housing and the role of the COVID-19 pandemic in both. Our study supports previous literature showing that Latinos tend to be over-represented in hazardous, low-wage occupations while being under-insured, resulting in an increased likelihood of experiencing work-related injuries and accidents and poorer health outcomes overall [37–39]. Our participants experienced job loss and subsequent loss of housing due to accidents and injuries that occurred on the job or on route to their place of employment. Although Latino/a employees were over-represented among frontline workers during the COVID-19 pandemic, we did not find that COVID-related illness contributed to job loss and subsequent homelessness [17]. Instead, our participants were particularly vulnerable to income loss resulting from COVID-related economic disruptions.

The rise in Latino/a homelessness calls for an increase in policy and programmatic attention. Homelessness and housing systems need to adapt to the changing demographic profile of the homeless population. Latino/a people experience various forms of discrimination, including barriers based on immigration status, concerns about the public charge, increased criminalization, and lack of responsiveness from the homelessness and housing systems. The Trump administration's reversal of a policy that previously restricted US Immigration and Customs Enforcement (ICE) operations in sensitive locations has increased barriers and risks for Latino/a people experiencing homelessness, particularly those who are undocumented. ICE has reportedly begun targeting homeless shelters, which may discourage Latino/a individuals from seeking shelter and support [40]. Strains on homelessness and housing systems across many US cities resulting from the recent influx of primarily Latino/a migrants heightens this urgency [18,41]. There is a need for increased enforcement of employment and fair housing laws, regardless of immigration status, and access to employment and housing benefits for those who qualify. The recent ruling in Grants Pass v Johnson that re-criminalized homelessness will only heighten criminalization of homelessness; our findings suggest that this will disproportionately fall on people of color and those who lack legal status, since these groups experience heightened criminalization and are vulnerable to having their civil rights violated [42]. Housing and homelessness service providers should increase culturally and linguistically tailored services and address the role that discrimination or other forms of unfair treatment may play in accessing these services.

Our study has several limitations. First, while the CASPEH study is representative of the Latino/a population in California, Latino/a in the state may differ from those in other parts of the U.S. in terms of legal status, country of origin, and cultural orientation, among other factors. Additionally, California policies related to housing and homelessness specifically may be unique and these results may not be representative of other regions. Furthermore, California's high housing costs

relative to wages play an outsize role in driving homelessness in the state; the experiences of homelessness among Latino/a residing in states with lower housing costs may be different from what we found. To build rapport and protect participants from unanticipated harm, we did not directly ask them about their immigration status, limiting our ability to draw comparisons and to fully understand how their immigration status impacted their experiences of homelessness. However, participants frequently disclosed their immigration status organically during the interview, which allowed us to take legal status into account during our subsequent analysis. Our study was conducted prior to an increase in migrant asylum-seekers from Latin America and other parts of the world who arrived at the US southern border in 2023 and 2024, whose numbers greatly impacted shelter and other housing services in several US cities. Further, our study took place prior to the Trump administration's second term, during which several policies have shifted in ways that negatively directly impact this population.

## Conclusion

The findings from this study highlight the complex and multifaceted challenges faced by Latino/a adults experiencing homelessness in California. These challenges stem from systemic discrimination, economic precarity, and cultural norms regarding responsibility, familial obligation, and personhood. These norms result in individual and interpersonal tensions between culturally prescribed notions of self-reliance and social support, leading people of Latino/a origin to experience homelessness and to underuse vital services. Further exacerbating these challenges are structural barriers, including language differences, immigration status, and discriminatory practices. While this study has provided valuable insight into the Latino/a experience of homelessness in California, additional research is needed to explore the effects of migration on the homelessness landscape, and the impacts of evolving homelessness and immigration policies. Only by initiating evidence-based measures that address the unique cultural, social, and structural conditions impacting unhoused and precariously housed Latino/a populations can we begin to address the increasing rates of homelessness among this population.

## Supporting Information

**S1 Data. Availability Statement for Publication_2.19.26.**
(DOC)

**S2 Data. Inclusivity in global research.**
(DOCX)

## Acknowledgments

We would like to thank our study participants for sharing their stories. We thank the study staff who conducted the research and the members of the community advisory boards who guided our work.

## Author contributions

**Conceptualization:** Michael Duke, Margot Kushel.

**Data curation:** Zena Kasandra Coronado, Michael Duke.

**Formal analysis:** Zena Kasandra Coronado, Michael Duke, Madison Rodriguez, Lourdes Johanna Avelar Portillo, Dafna Erana Hernandez.

**Funding acquisition:** Margot Kushel.

**Investigation:** Zena Kasandra Coronado, Michael Duke, Madison Rodriguez, Margot Kushel.

**Methodology:** Zena Kasandra Coronado, Michael Duke, Margot Kushel.

**Project administration:** Zena Kasandra Coronado, Michael Duke, Margot Kushel.

**Resources:** Margot Kushel.

**Software:** Zena Kasandra Coronado, Michael Duke.

**Supervision:** Michael Duke, Margot Kushel.

**Writing – original draft:** Zena Kasandra Coronado, Michael Duke, Madison Rodriguez, Lourdes Johanna Avelar Portillo, Dafna Erana Hernandez.

**Writing – review & editing:** Margot Kushel.

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
