## [Decision Letter · Decision Letter 0]

28 Jul 2025

Dear Dr. Kushel,

Thank you for submitting your manuscript to PLOS ONE. After careful consideration, we feel that it has merit but does not fully meet PLOS ONE’s publication criteria as it currently stands. Therefore, we invite you to submit a revised version of the manuscript that addresses the points raised during the review process.

We look forward to receiving your revised manuscript.

Kind regards,

Kimberly Page, PhD, MPH

Academic Editor

PLOS ONE

Journal Requirements:

2. (1) In the ethics statement in the methods, you have specified that verbal consent was obtained. Please provide additional details regarding how this consent was documented and witnessed, and state whether this was approved by the IRB.

(2) Please include a complete copy of PLOS’ questionnaire on inclusivity in global research in your revised manuscript. Our policy for research in this area aims to improve transparency in the reporting of research performed outside of researchers’ own country or community. The policy applies to researchers who have travelled to a different country to conduct research, research with Indigenous populations or their lands, and research on cultural artefacts. The questionnaire can also be requested at the journal’s discretion for any other submissions, even if these conditions are not met.  Please find more information on the policy and a link to download a blank copy of the questionnaire here: https://journals.plos.org/plosone/s/best-practices-in-research-reporting. Please upload a completed version of your questionnaire as Supporting Information when you resubmit your manuscript.”

4. We note you have included a table to which you do not refer in the text of your manuscript. Please ensure that you refer to Table 1 in your text; if accepted, production will need this reference to link the reader to the Table.

5.If the reviewer comments include a recommendation to cite specific previously published works, please review and evaluate these publications to determine whether they are relevant and should be cited. There is no requirement to cite these works unless the editor has indicated otherwise.

Additional Editor Comments:

Both reviewers point out areas for manuscript improvement.  I agree with them that this research is valuable and highly needed. Given the complicated environment for health research in this area and potential new policies, I expect that this article will be highly cited as many agencies and communities look for support and compassionate approaches to support Hispanic people experiencing homelessness.  I do note that neither reviewer commented on the term Latinx.  I understand the effort to find a term that is applicable to the many groups of Hispanic/ who are in our communities.  The term Latinx is not widely used and may puzzle some readers who are less academic. (I use the term Hispanic because I am in New Mexico and that is the normal term used by most in our area. California context should be considered for the terminology you use and not me or my context!)

Reviewers' comments:

Reviewer's Responses to Questions

**Comments to the Author**

1. Is the manuscript technically sound, and do the data support the conclusions?

Reviewer #1: Yes

Reviewer #2: Yes

2. Has the statistical analysis been performed appropriately and rigorously?

Reviewer #1: Yes

Reviewer #2: N/A

3. Have the authors made all data underlying the findings in their manuscript fully available?

Reviewer #1: Yes

Reviewer #2: No

4. Is the manuscript presented in an intelligible fashion and written in standard English?

Reviewer #1: Yes

Reviewer #2: Yes

Reviewer #1: This is a well done and informative study to identify and describe drivers of Latino/a homeless in California (CA). Qualitative interviews of 84 participants from the larger CASPEH study provide the data. The introduction makes a compelling case for Latino/a homelessness as a growing public health problem. Authors do a good job laying out challenges to sampling CA Latino/as in page 4 and justification for venue based sampling supplemented by respondent driven sampling.

More detail on how “purposive sampling” which should be defined (lines 127-128) is used to choose 365 participants for qualitative interviews from the 3,200 is needed. Authors should address “purposive sampling” not leading to bias. The sample for this manuscript (Lines 141-143) comes from Latino/as from one of 7 interview based sub-studies. Some basic demographic information about similarities and differences of Latinos in these different sub-studies would be helpful. For example, what percentage of interviews were conducted in Spanish in each of the seven sub-studies as a marker of acculturation, age, gender, birth outside US?

The term “housing returns” in abstract line 45 and line 125 should be defined. Perhaps this means an inability to return to temporary housing?

The interview quotes in the Results section are illustrative and support the themes under which they are grouped.

Lines 459-461- I would add word “negatively”: “Our study took place prior to the Trump administration’s second term, during which several policies have shifted in ways that negatively directly impact this population.”

This is a timely qualitative study with valuable difficult to obtain interview data on drivers of Latino/a homelessness in the US’ most populous state. It is an important contribution to the public health and social science literature.

Reviewer #2: This article addresses a highly relevant topic that fills a critical gap in our understanding of homelessness experiences. While homelessness research has expanded significantly in recent decades, studies specifically examining the experiences of Latina/o/e populations remain notably underrepresented in the literature. This gap is particularly concerning given the documented inequities in housing insecurity and the unique cultural, linguistic, and socioeconomic factors that may influence homelessness trajectories within these communities. Furthermore, California's significant Latina/o/e population, relative high rates of homelessness, and dedicated resources to address it, make this research especially important. Approaches like the one employed in this research are both welcome and urgently needed to advance our understanding of how structural factors intersect with homelessness experiences. I recommend the article for publication with one minor suggestion: The established homelessness research literature indicates that interviews with people experiencing homelessness typically encompass healthcare access and utilization, mental and behavioral health topics, and other aspects of the health-illness continuum, in addition to the themes examined in this study. I suggest that the authors acknowledge in their methodology section which topics were intentionally excluded from data collection and/or analysis and provide a rationale for these decisions on the scope of the study. This would help readers better understand the authors' criteria for topic inclusion and exclusion, thereby strengthening the methodological strengths of the work.

**Do you want your identity to be public for this peer review?** For information about this choice, including consent withdrawal, please see our Privacy Policy

Reviewer #1: No

Reviewer #2: **Yes:** Laura Nervi

---

## [Author Response · Author response to Decision Letter 1]

12 Feb 2026

Dear Dr. Page,

Thank you and the reviewers for your thoughtful and encouraging feedback on our manuscript, “Latino/a experiences of homelessness in California: Qualitative findings from the California Statewide Study of People Experiencing Homelessness (CASPEH).” We appreciate the recognition that our manuscript is “valuable and highly needed.” We are grateful for the acknowledgment that our manuscript will be “highly cited as many agencies and communities look for support and compassionate approaches to support Hispanic people experiencing homelessness.”

Below are our responses to the reviewers’ comments and actions to address them in the paper revision.

Journal Requirements:

Author response: We reviewed the journal’s style requirements and made edits accordingly.

2. In the ethics statement in the methods, you have specified that verbal consent was obtained. Please provide additional details regarding how this consent was documented and witnessed, and state whether this was approved by the IRB.

Author response: After discussion with the UCSF IRB who approved all parts of the California State Study of People Experiencing Homelessness (CASPEH), we chose to create verbal consent processes for both the survey and related in-depth interviews. Because we completed the study anonymously (we did not collect participants’ names or other identifying information), the IRB noted that collecting signed consents could increase the participants’ risk, as we would have their name (on the consent). Thus, we created a two-step verbal consent process for the survey (first, a pre-screen to see if the potential participant consented to be asked screening questions, and then if they were eligible, a longer verbal consent. The trained staff read a script explaining the study, including potential risks. Participants were told that they could stop at any time. Prior to giving verbal consent, the participant had to briefly explain their understanding of the study and its risks and benefits to the staff member. If they were able to do this, the participant completed a verbal consent, asserting that they wanted to be in the study. If they did so, the trained staff recorded their assent into the REDCap screening tool, which then assigned the participant a unique ID code. If the participant was unable to briefly explain the study and the risks, confirming that they could consent, the staff member repeated the information. If the participant was unable to do so, the participant thanked them for their time and marked them as unable to give consent. Only participants who had completed the administered questionnaire were eligible for the in-depth interview study, from which we drew the data for this paper.

At the completion of the administered survey, the staff member who completed the survey would (if the participant were eligible) ask them if they would like to be considered for the in-depth interview. If the participant expressed interest, they were offered a break if they wished. If they wanted to proceed, a separate staff member approached and completed another verbal consent for the qualitative study using a similar process. If the participant consented, the in-depth interview staff member would record their assent.

The consent process was developed with recommendations from the UCSF Institutional Review Board and approved by them.

3. Please include a complete copy of PLOS’ questionnaire on inclusivity in global research in your revised manuscript. Our policy for research in this area aims to improve transparency in the reporting of research performed outside of researchers’ own country or community. The policy applies to researchers who have travelled to a different country to conduct research, research with Indigenous populations or their lands, and research on cultural artefacts. The questionnaire can also be requested at the journal’s discretion for any other submissions, even if these conditions are not met. Please find more information on the policy and a link to download a blank copy of the questionnaire here: https://journals.plos.org/plosone/s/best-practices-in-research-reporting. Please upload a completed version of your questionnaire as Supporting Information when you resubmit your manuscript.”

Author response: We have completed the questionnaire and uploaded it as part of our resubmission.

Author response: Because of the high likelihood of respondents describing illegal or socially-sanctioned behaviors in specific community settings, coupled with the current targeting of immigrant populations for deportation, our Lived Expertise Advisory Board urged us to maintain control of the de-identified data. While we collected the data anonymously, the detailed data given by participants in in-depth interviews (which often included information such as documentation status, location of interview, and detailed information that could possibly be identifying), the Community Advisory Board asked us to exercise caution. In light of these factors, and in consultation with UCSF’s Director of Data Science & Open Scholarship, we agree to share the study data based on individual requests to Margot Kushel, the study PI (Margot.Kushel@uscf.edu). Dr. Kushel will discuss data requests and ensure that data shared does not contain potentially identifiable information. Please note that UCSF does not currently have a data access committee.

5. We note you have included a table to which you do not refer in the text of your manuscript. Please ensure that you refer to Table 1 in your text; if accepted, production will need this reference to link the reader to the Table.

Author response: Thank you for identifying this error. We have added a reference to the table in line 146.

Reviewer #1

1. This is a well done and informative study to identify and describe drivers of Latino/a homeless in California (CA). Qualitative interviews of 84 participants from the larger CASPEH study provide the data. The introduction makes a compelling case for Latino/a homelessness as a growing public health problem. Authors do a good job laying out challenges to sampling CA Latino/as in page 4 and justification for venue based sampling supplemented by respondent driven sampling.

Author response: We thank the reviewer for their positive feedback about our manuscript.

2. More detail on how “purposive sampling” which should be defined (lines 127-128) is used to choose 365 participants for qualitative interviews from the 3,200 is needed. Authors should address “purposive sampling” not leading to bias.

Author response: We have added the following verbiage to clarify use of purposive sampling in the study (lines 127-134): “We used purposive sampling to select participants for the qualitative interviews (N = 365). Purposive sampling is a technique in qualitative research to identify participants who have information, experiences, or characteristics that are relevant to the aims of the study. Participants were selected based on a) having completed the survey b) being flagged for a particular qualitative sub-study based on their responses to specific survey questions, and c) the survey interviewer’s assessment of the participant’s ability to discuss the topic at length.”

3. The sample for this manuscript (Lines 141-143) comes from Latino/as from one of 7 interview based sub-studies. Some basic demographic information about similarities and differences of Latinos in these different sub-studies would be helpful. For example, what percentage of interviews were conducted in Spanish in each of the seven sub-studies as a marker of acculturation, age, gender, birth outside US?

Author response: The sample for this manuscript comes from participants who identified as Latino/a across the seven sub-studies, not just one. Demographic information about these participants is included in Table 1. We have added reference to the table in line 146 to point readers to this demographic information.

4. The term “housing returns” in abstract line 45 and line 125 should be defined. Perhaps this means an inability to return to temporary housing?

Author response: We use the term “housing returns” to mean re-entering housing so that the participant no longer meets the definition of homelessness. This would include moving in with family so long as the assumption was that the participant could stay as long as they wanted, moving into one’s own room or apartment, or moving into housing with a permanent subsidy. We recognize that there is never a guarantee that people are able to remain in housing long-term, but we focused on returns to housing with enough stability that people no longer met formal criteria for homelessness.

We have defined this term in lines 46 – 49.

5. The interview quotes in the Results section are illustrative and support the themes under which they are grouped.

Author response: Thank you for this positive feedback. We agree that the quotes were clear and effectively support the analysis.

6. Lines 459-461- I would add word “negatively”: “Our study took place prior to the Trump administration’s second term, during which several policies have shifted in ways that negatively directly impact this population.”

Author response: We appreciate this suggestion and have added the word “negatively” to line 469.

7. This is a timely qualitative study with valuable difficult to obtain interview data on drivers of Latino/a homelessness in the US’ most populous state. It is an important contribution to the public health and social science literature.

Author response: We appreciate the reviewer’s encouraging feedback. We are pleased that they recognize the significance and timeliness of this work, as well as the value of the perspectives of our participants. Thank you for acknowledging its contribution to public health and social science literature.

Reviewer #2

1. This article addresses a highly relevant topic that fills a critical gap in our understanding of homelessness experiences. While homelessness research has expanded significantly in recent decades, studies specifically examining the experiences of Latina/o/e populations remain notably underrepresented in the literature. This gap is particularly concerning given the documented inequities in housing insecurity and the unique cultural, linguistic, and socioeconomic factors that may influence homelessness trajectories within these communities. Furthermore, California's significant Latina/o/e population, relative high rates of homelessness, and dedicated resources to address it, make this research especially important. Approaches like the one employed in this research are both welcome and urgently needed to advance our understanding of how structural factors intersect with homelessness experiences.

Author response: We are grateful for the reviewer’s thoughtful and supportive comments. We agree that the underrepresentation of Latino/a experiences in homelessness research is a critical gap, especially given the structural inequities affecting these communities. We hope this study contributes meaningfully to advancing equitable and culturally informed responses to homelessness.

2. I recommend the article for publication with one minor suggestion: The established homelessness research literature indicates that interviews with people experiencing homelessness typically encompass healthcare access and utilization, mental and behavioral health topics, and other aspects of the health-illness continuum, in addition to the themes examined in this study. I suggest that the authors acknowledge in their methodology section which topics were intentionally excluded from data collection and/or analysis and provide a rationale for these decisions on the scope of the study. This would help readers better understand the authors' criteria for topic inclusion and exclusion, thereby strengthening the methodological strengths of the work.

Author response: We viewed a specific focus on the physical and mental health of the study population as beyond the scope of this manuscript. We appreciate the comment and note that we did extensive data collection in the survey on these topics, which we have published separately. The Latino/x interview guide did not focus on these questions, choosing instead to focus on the lived experience of Latino/x participants facing homelessness, although other in-depth interview guides did.

We have added a note that we excluded these topics from this analysis (lines 135-139).

Editor:

1. I do note that neither reviewer commented on the term Latinx. I understand the effort to find a term that is applicable to the many groups of Hispanic/ who are in our communities. The term Latinx is not widely used and may puzzle some readers who are less academic. (I use the term Hispanic because I am in New Mexico and that is the normal term used by most in our area. California context should be considered for the terminology you use and not me or my context!)

Author response: We use the term Latino/a to refer to our participants who indicated Latino/a/x, Hispanic, or Latin American on the CASPEH race measure. We recognize there is an ongoing debate around the most appropriate terminology to use for this population. We chose not to use Latinx because, as you mentioned, it is not widely adopted outside of academic settings. We opted against Hispanic, given its associations with Spain and colonialism, which may alienate individuals who do not identify with the Spanish language or heritage (e.g., those who speak indigenous languages). Our use of the term Latino/a aims to reflect a term that is more broadly familiar.

---

## [Editor Report · Decision Letter 1]

15 Feb 2026

Latino/a experiences of homelessness in California: Qualitative findings from the California Statewide Study of People Experiencing Homelessness (CASPEH)

PONE-D-25-27066R1

Dear Dr. Kushel,

We’re pleased to inform you that your manuscript has been judged scientifically suitable for publication and will be formally accepted for publication once it meets all outstanding technical requirements.

Kind regards,

Kimberly Page, PhD, MPH

Academic Editor

PLOS One

Additional Editor Comments (optional):

I am very happy to endorse publication of this paper.
---

## [Editor Report · Acceptance letter]

PONE-D-25-27066R1

PLOS One

Dear Dr. Kushel,

I'm pleased to inform you that your manuscript has been deemed suitable for publication in PLOS One. Congratulations! Your manuscript is now being handed over to our production team.

Kind regards,

on behalf of

Dr. Kimberly Page

Academic Editor

PLOS One